# The Relationship of Energy Malnutrition, Skeletal Muscle and Physical Functional Performance in Patients with Stable Chronic Obstructive Pulmonary Disease

**DOI:** 10.3390/nu14132596

**Published:** 2022-06-23

**Authors:** Manabu Tomita, Masaru Uchida, Yujiro Imaizumi, Megumi Monji, Emiko Tokushima, Michihiro Kawashima

**Affiliations:** 1Graduate School of Medicine, Kurume University, Kurume 830-0011, Japan; 2Department of Rehabilitation, Japan Community Health Care Organization Saga Central Hospital, Saga 849-8522, Japan; yy.imaizumi@gmail.com; 3Department of Respiratory Medicine, Japan Community Health Care Organization Saga Central Hospital, Saga 849-8522, Japan; uchidamer@gmail.com (M.U.); megumi.m.jchosaga@gmail.com (M.M.); emikotokushima@hotmail.com (E.T.); michihirok@hotmail.com (M.K.)

**Keywords:** chronic obstructive pulmonary disease, energy malnutrition, erector spinae muscle, indirect calorimetry, physical functional performance, respiratory quotient, ventilation volume

## Abstract

Weight loss is a factor that affects prognosis in patients with chronic obstructive pulmonary disease (COPD) independent of lung function. One of the major factors for weight loss is energy malnutrition. There have been no reports on the factors related to energy malnutrition in COPD patients. This retrospective observational study aimed to investigate these factors. We included 163 male subjects with COPD. Respiratory quotient (RQ), an index of energy malnutrition, was calculated by expiratory gas analysis using an indirect calorimeter. RQ < 0.85 was defined as the energy-malnutrition group and RQ ≥ 0.85 as the no energy-malnutrition group. Factors related to energy malnutrition were examined by multivariate and decision-tree analysis. We finally analyzed data from 56 selected subjects (median age: 74 years, BMI: 22.5 kg/m^2^)**.** Energy malnutrition was observed in 43%. The independent factors associated with energy malnutrition were tidal volume (VT) (OR 0.99; 95% CI 0.985–0.998; *p* = 0.015) and Th12 erector spinae muscle cross-sectional area SMI (Th12ESM_SMI_) (OR 0.71; 95% CI 0.535–0.946; *p* = 0.019). In decision-tree profiling of energy malnutrition, VT was extracted as the first distinguishable factor, and Th12ESM_SMI_ as the second. In ROC analysis, VT < 647 mL (AUC, 0.72) or Th12ESM_SMI_ < 10.1 (AUC, 0.70) was the cutoff value for energy malnutrition. Energy malnutrition may be an early warning sign of nutritional disorders.

## 1. Introduction

In Japan, about 70% of patients with chronic obstructive pulmonary disease (COPD) have a %ideal body weight (%IBW) of less than 90%, and the frequency of nutritional disorders is higher than in Europe and the United States [1,2,3,4]. Malnutrition in COPD patients is associated with cachexia, sarcopenia, and weight loss and may increase the risk of decreased lung function and decreased exercise capacity [5]. Weight loss is a prognostic factor for COPD patients independent of lung function [6,7]. The cause of weight loss is increased muscle protein catabolism due to energy shortage caused by decreased energy intake and increased resting energy expenditure caused by effortful breathing and inflammatory cytokines [8]. It has been reported that nutritional therapy is indicated when %IBW < 90% [9], and aggressive nutritional support therapy is necessary when %IBW < 80% [10]. Thus, weight loss is a significant risk factor in COPD patients.

In general, one of the major factors for weight loss is starvation or energy malnutrition. It was reported that weight loss in COPD was mainly due to inadequate nutritional intake [11]. Among malnutrition, the state in which the body lacks the energy to move is called energy malnutrition and is generally referred to as starvation [12]. In a state of starvation, hepatic glycogen is broken down to supply glucose, but the amount of glucose stored is small and is depleted within a few hours [13]. In addition, skeletal muscle mass decreases from 3 to 4 days after starvation because energy is produced by freeing protein and fat from muscles [14]. Therefore, patients who have a long fasting time at night may suffer from short-term starvation due to glycogen depletion, resulting in a reduction in skeletal muscle mass.

In elderly COPD patients, in addition to decreased energy intake and increased resting energy expenditure due to effortful breathing and inflammatory cytokines, there is often a long fasting time between dinner and breakfast. This means that there is a possibility of starvation, or energy malnutrition, in the early morning. Determination of early morning starvation or energy malnutrition and examination of its associated factors will contribute to the early detection and prevention of malnutrition and weight loss. However, there have been no reports on factors related to energy malnutrition in COPD patients. In this study, we used an indirect calorimeter to measure metabolism during early morning fasting. We calculated the respiratory quotient (RQ), which is an index of energy malnutrition, and investigated the actual state of energy malnutrition in patients with stable COPD. Furthermore, we examined the relationship between respiratory function, biochemical blood tests, muscle mass, and physical function.

## 2. Methods

### 2.1. Study Design and Subjects

This study was a retrospective observational study designed to investigate the factors associated with energy malnutrition in male outpatients with COPD. From October 2020 to March 2021, a total of 163 male outpatients with COPD attended Japan Community Health Care Organization Saga Central Hospital. Inclusion criteria: male outpatients with COPD who (1) were 60 years of age or older, (2) able to perform all physical function tests, and (3) had undergone a biochemical examination and CT scans for Th12 level.

### 2.2. Measurement of Energy Metabolism

Energy metabolism was measured with the use of an indirect calorimeter (Aeromonitor AE-300S; Minato Medical Science, Tokyo, Japan). Subjects were not to eat or drink anything after 9:00 p.m. on the day before the measurement, except for the consumption of calorie-free water or tea. In addition, the amount of food eaten on the day before the measurement was questioned to confirm. There were no subjects who had insufficient food intake. All measurements were taken at 9:30 a.m. in a fasted state after bed rest for at least 30 min as in previous studies [15]. Oxygen consumption (VO_2_) and carbon dioxide production (VCO_2_) were measured for 15 min, and the average of the last 10 min was used for analysis [16]. The measured VO_2_ and VCO_2_ were then used to calculate the RQ for each patient (RQ = VCO_2_/VO_2_). The RQ is a dimensionless number used in calculations of basal metabolic rate (BMR) when estimated from carbon dioxide production. It is calculated from the ratio of carbon dioxide produced by the body to oxygen consumed by the body. The respiratory quotient value indicates which macronutrients are being metabolized, as different energy pathways are used for fats, carbohydrates, and proteins [17]. In addition, resting energy expenditure (REE), respiratory rate (RR), tidal volume (VT), minute ventilation (VE), carbohydrate oxidation rate, and fat oxidation rate were measured. The carbohydrate oxidation rate and fat oxidation rate were calculated using Elwyn’s energy calculation formula [18]. Basal energy expenditure (BEE) was predicted with the Harris–Benedict equation.

### 2.3. Diagnosis of Energy Malnutrition

In general, the turning point for the substrate of thermogenesis from carbohydrate dominant (npRQ ≥ 0.85) towards fat dominant (npRQ < 0.85) is 0.85 of non-protein RQ (npRQ) [19]. In addition, McClave SA et al. defined RQ lower than 0.85 as underfeeding and higher than 1.0 as overfeeding in the measured RQ [20]. Furthermore, liver disease indicates starvation when RQ is less than 0.85 due to insufficient glycogen accumulation in the liver; it has been reported that npRQ < 0.85 is the best cutoff value for energy malnutrition [21]. With these references, we defined RQ < 0.85 as energy malnutrition (underfeeding) and RQ ≥ 0.85 as no energy malnutrition in this study.

### 2.4. Anthropometry

Before measuring energy metabolism, anthropometry measurement was performed to determine body weight, body mass index (BMI), body fat, and skeletal muscle mass using bioelectrical impedance analysis (MC-780MA-N; Tanita; Tokyo, Japan). Skeletal muscle mass was normalized by the square of height, and the data were expressed as skeletal muscle mass index (SMI).

### 2.5. Lung Function

Spirometry was performed by three well-trained pulmonary technologists according to the manual of the American Thoracic Society/European Respiratory Society Task Force, using the same spirometer (Autospiro AS-507, Minato Medical Science, Osaka, Japan) for all participants [22]. The forced expiratory volume in 1 s (FEV1), forced vital capacity (FVC), and FEV1/FVC ratio was obtained.

### 2.6. Laboratory Determinations

Venous blood was collected in the morning after 12 h of fasting. To examine liver function, lipid metabolism, glucose metabolism, and muscle metabolism, which may affect starvation, before measuring energy metabolism, blood samples were taken and analyzed for these parameters: albumin (Alb), transthyretin (TTR), C-reactive protein (CRP), aspartate aminotransferase (AST), alanine aminotransferase (ALT), gamma-glutamyl transpeptidase (GGT), total cholesterol (T-Cho), HDL cholesterol (HDL-C), LDL cholesterol (LDL-C), triglyceride (TG), creatinine (Cre), and blood glucose (BG).

### 2.7. Evaluation of Skeletal Muscle Mass

Skeletal muscle mass was calculated by normalization with a CT scan (Revolution EVO; GE Healthcare, Madison, WI, USA) at the level of the 12th thoracic vertebra (Th12). In this study, the cross-sectional area of total skeletal muscle and erector spinae muscle (ESM) at the Th12 level were selected [23]. This analysis was performed using a SYNAPSE VINCENT volume analyzer (FUJIFILM Medical, Tokyo, Japan). The cross-sectional muscle area was normalized by the squared height and represented as SMI.

### 2.8. Evaluation of Physical Function, Activities of Daily Living (ADL), and Nutrition Status

Grip strength was measured using a hand dynamometer (Digital Grip Dynamometer, Takei Scientific Instruments, Niigata, Japan). Maximum grip strength was measured three times with the dominant hand, and the maximum value was used as the measurement value. Knee extension strength was determined using a hand-held dynamometer (μTas MT-1; Anima, Tokyo, Japan). Subjects were instructed to press the dynamometer to straighten their knees. They then maintained their max efforts for an extra 5 s. Gait speed (m/s) was calculated by the 10 m walking test (10MWT) [24]. The 10MWT was performed twice, and the faster one was used for analysis. The Timed Up and Go Test (TUG) is to measure the time required for a patient to rise from an armless chair, walk a three-meter distance, turn around, return to the chair, and take a seat [25]. The test was performed twice at maximum speed, and the fastest result was selected for analysis. The 5-time chair stand test (CS-5) was performed according to the Asian Working Group for Sarcopenia protocol. The test was evaluated by the time required to stand 5 times from a sitting position while arms were folded across the chest [26]. The test was performed once at maximum effort. The 6 min walk test (6MWT) was measured as described in the American Thoracic Society (ATS) guidelines [27]. The maximum walking distance was used for analysis. The Nagasaki University Respiratory ADL questionnaire (NRADL) was used to assess activities of daily living [28]. The Mini Nutritional Assessment^®^-short form (MNA-SF) assessed nutritional status. Subjects whose total score < 12 were categorized as at risk for malnutrition, while those whose total score ≥ 12 were categorized as having normal nutrition status [29].

### 2.9. Statistical Analysis

Data were represented as median (interquartile range (IQR)), range, or number. Values for differences between the energy malnutrition and non-energy malnutrition groups were analyzed by use of Wilcoxon rank-sum tests. In addition, logistic regression analysis was used to analyze the independent factors related to energy malnutrition. For the selection of candidates for logistic regression analysis, we used the single factor regression analysis (Spearman’s tests). Factors associated with energy malnutrition were profiled using decision tree analysis. A decision-tree algorithm is a data-mining technique that reveals a series of classification rules by identifying priorities and therefore allows clinicians to choose an option that maximizes the benefit for the patient [30]. Decision trees are a popular modeling technique in economics and clinical practice and have proved their usefulness in human medicine [31,32,33]. Finally, the Receiver Operating Characteristic (ROC) curve using the Youden index was used to determine the best cutoff value of independent factors to discriminate RQ < 0.85. All statistical analyses were conducted using Statistical Analysis Software (JMP Pro version 15.0; SAS Institute, Cary, NC, USA). The statistical significance level was set at *p* < 0.05.

## 3. Results

We finally analyzed data from 56 selected subjects who fulfilled the criteria for exclusion as follows: outpatients with COPD who (1) had severe liver malfunction; (2) had severe diabetes; (3) suffered from inflammatory, endocrine, or gastrointestinal diseases; (4) had other malignancies, malabsorption, or motor disorders; (5) were using home oxygen (because the RQ cannot be accurately measured) (Figure 1).

### 3.1. Subjects’ Characteristics

The characteristics of the subjects are listed in Table 1. The subjects had a median age of 74 years, 100% of whom were male, and a median BMI of 22.5 kg/m^2^. Subjects with GOLD stage I, II, III, and IV were 21.4%, 41.0%, 32.1%, and 5.3%, respectively. Subjects with mMRC dyspnea scale score 0, 1, 2, 3 and 4 were 25.0%, 46.4%, 14.2%, 10.7% and 3.5%, respectively. The median MNA-SF was 12 points, and the nutritional status was comparably normal. The median NRADL score was 95, and the median patient was almost independent in ADL. The median Alb, total TTR, and CRP were 4.2 g/dL, 28.3 mg/dL, and 0.09 mg/dL, respectively. The median T-Cho, Cre and BG were 197.5 mg/dL, 0.87 mg/dL and 103.0 mg/dL, respectively. The median RQ was 0.85, and the median energy oxidation rate was 50.8% for carbohydrates and 49.2% for fats (Table 1).

### 3.2. Comparison of Baseline, Energy Metabolism, Biochemical Tests, Body Composition, Physical Function, and Muscle Mass between Subjects with and without Energy Malnutrition (RQ < 0.85 and RQ ≥ 0.85)

In univariate analysis, no significant difference was found in age, BMI, %IBW, GOLD stage, the severity of dyspnea, and lung function between the with and without energy malnutrition groups. Subjects in the energy malnutrition (RQ < 0.85) group had significantly lower carbohydrate oxidation rate, had significantly higher fat oxidation rate, and had significantly lower VT than subjects in the non-energy malnutrition group (RQ ≥ 0.85). However, there was not any significant difference in REE, RR, or VE between the two groups. Subjects in the energy malnutrition (RQ < 0.85) group had significantly lower MNA-SF scores and had significantly lower Th12 muscle mass, Th12ESM mass, and Th12ESM_SMI_ than subjects in the non-energy malnutrition group (RQ ≥ 0.85). Otherwise, there were no significant differences in biochemical tests, body composition, or physical function between the two groups (Table 2).

### 3.3. Logistic Regression Analysis and Decision-Tree Analysis for Energy Malnutrition

The results of single regression analysis (spearman’s tests) between RQ and other parameters showed that VT (*r* = 0.349, *p* = 0.0084), Th12ESM_SMI_ (*r* = 0.390, *p* = 0.0035) and MNA-SF (*r* = 0.360, *p* = 0.0063) were significantly correlated. In a logistic regression analysis with VT, Th12ESM_SMI_, and MNA-SF as explanatory variables and energy malnutrition as an objective variable, the independent negative risk factors for energy malnutrition were both VT (OR 0.99; 95% CI 0.984–0.998; *p* = 0.010) and Th12ESM_SMI_ (OR 0.73; 95% CI 0.517–0.978; *p* = 0.034) (Table 3). The decision-tree analysis revealed that the following three profiles were related to energy malnutrition: Profile 1) VT < 701 mL, Profile 2) Th12ESM_SMI_ < 9.59, and Profile 3) BMI < 22.6 and %FEV1 < 58.2% (Figure 2).

### 3.4. VT and Th12 ESM_SMI_ for Discrimination of Energy Malnutrition

The ROC curve was analyzed to establish the optimal cutoff value for VT or Th12ESM_SMI_ for distinguishing energy malnutrition. When the VT cutoff value was set at 647 mL, the area under the curve (AUC) was 0.72, and the sensitivity and specificity were 87% and 50%, respectively. Furthermore, when the cutoff value of Th12ESM_SMI_ was set at 10.1, the AUC was 0.70, with a sensitivity of 65% and specificity of 74% (Figure 3).

## 4. Discussion

This study is the first to investigate the factors related to energy malnutrition in stable COPD subjects by calculating the RQ through exhaled gas analysis using an indirect calorimeter. We have shown that energy malnutrition was seen in 43% of subjects with stable COPD. The independent factors associated with energy malnutrition were VT and Th12ESM_SMI_.

In the patient background, 43% had energy malnutrition. This indicates that 43% of the subjects metabolize fat significantly more than carbohydrates, which may indicate a shift toward weight loss. It was reported that weight loss with a BMI of less than 20 kg/m^2^ was seen in about 30% of patients, even though patients with mild to moderate disease accounted for about 70% [34]. In addition, it has been reported that a decrease in fat mass can be observed in mild weight loss (80% ≤ %IBW < 90%), and a decrease in lean mass can be observed in moderate or greater weight loss (%IBW < 80%) [4]. In the current study, 43% of the subjects had energy malnutrition, although the severity of the disease was about the same, and the %IBW was greater than in previous reports. This may suggest that the signs of weight loss may have been caught early.

In a logistic regression analysis, VT was identified to be a risk factor for energy malnutrition (RQ < 0.85). Regarding the relationship between VT and malnutrition, Arora NS et al. reported that malnutrition decreases both respiratory muscle strength and maximal spontaneous ventilation, possibly impairing respiratory muscle strength to cope with the increased ventilatory load in thoraco-pulmonary disease [35]. In addition, Yoneda T et al. reported that the degree of airway obstruction and respiratory muscle function was associated with malnutrition characterized by the reduction of the Fischer ratio [36]. Taken together, although we did not measure respiratory muscle strength in this study, we hypothesize that the decrease in VT was due to the decrease in respiratory muscle strength caused by energy malnutrition. However, lung hyperinflation may also be a factor in the decline in VT. Although no index of lung hyperinflation has been measured in this study, lung hyperinflation is highly dependent on the degree of breathlessness [37]. In this study, there was no significant difference in the mMRC dyspnea scale score between the energy malnutrition and no energy malnutrition groups. Therefore, we hypothesized that the effect of lung hyperinflation may be small. We also discussed the association between VT and RQ. There was no significant increase in RR in the energy malnutrition group. However, the decrease in VT may cause inadequate expiration, and RR may compensate for this condition.

In addition, logistic regression analysis identified Th12ESM_SMI_ as a risk factor for energy malnutrition. Nutritional impairment in COPD patients has been reported to cause a high rate of lean body mass loss [1]. However, in this study, there was no significant difference in lean body mass measured by the BIA and a significant difference in ESM muscle mass on CT in the energy malnutrition group compared to the non-energy malnutrition group, and there was. It has been reported that ESM cross-sectional area by chest CT correlates with COPD clinical parameters and is a strong risk factor for overall mortality in COPD patients [23,38,39,40]. The relationship between ESM muscle mass and nutritional status is still unclear. However, in our study, ESM muscle mass was associated with energy malnutrition, suggesting that a decrease in ESM muscle mass may be associated with the early state of nutritional disorders.

As a result of the profiling of energy malnutrition by decision tree analysis, the group with VT less than 701 mL, Th12ESM_SMI_ less than 9.59, and %FEV1 less than 58.2% had 100% energy malnutrition (Figure 2). All of these factors are informed by daily clinical practice and may indicate the need for early nutritional intervention in patients who match the previously described criteria. These indices may be useful as indicators of the risk of energy malnutrition during pulmonary function tests and chest computed tomography in clinical practice.

In the ROC analysis, the cutoff values for energy malnutrition were VT < 647 mL (AUC, 0.72) or Th12 ESM_SMI_ < 10.1 (AUC, 0.70) (Figure 3). Both VT and Th12 ESM_SMI_ showed AUC ≥ 0.7, indicating moderate accuracy. These indices may be useful as indicators of the risk of energy malnutrition during pulmonary function tests and chest computed tomography in clinical practice.

This retrospective observational study has several limitations. First, the number of subjects was too small to allow for sufficient multivariate analysis. Second, the study was conducted at a single center, which may have led to various selection biases. Third, subjects on home oxygen were excluded from the study because of metabolic measurements by indirect calorimetry, and subjects with severe disease were not included. Therefore, further studies designed as prospective, multicenter studies in COPD patients with various conditions are required to explore the impact of energy malnutrition on skeletal muscle and physical functional performance in COPD patients. Moreover, as for metabolic measurements, we could not measure non-protein RQ because of the lack of 24-h urine storage in outpatients. Therefore, we could not take into account the energy metabolism of protein.

## 5. Conclusions

In conclusion, we have shown that 43% of the subjects with stable COPD had energy malnutrition, suggesting the possibility of early nutritional impairment and the need for early nutritional intervention. Furthermore, the factors associated with energy malnutrition were VT and Th12ESM_SMI_. Energy malnutrition may be an early warning sign of nutritional disorders. Taken together, it is beneficial to be able to predict energy malnutrition using clinical indicators.

## Figures and Tables

**Figure 1 nutrients-14-02596-f001:**
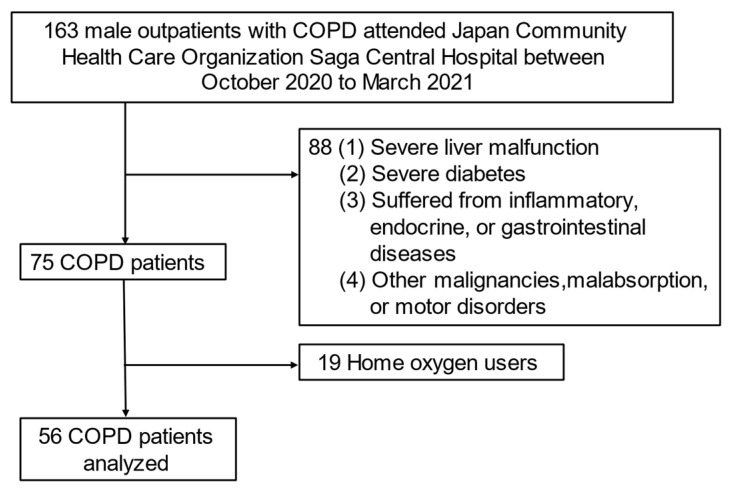
Flowchart of subject recruitment in this study. Abbreviation: COPD—chronic obstructive pulmonary disease.

**Figure 2 nutrients-14-02596-f002:**
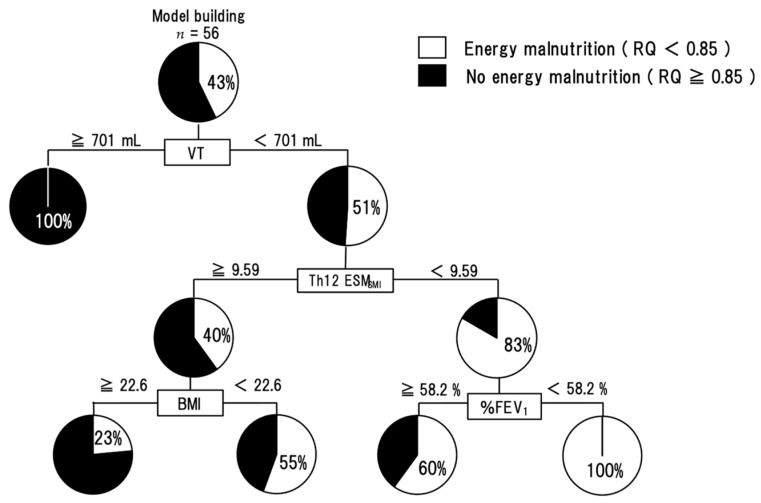
A decision-tree analysis for energy malnutrition. The pie graphs indicate the proportion of subjects with RQ < 0.85 (white) and subjects with RQ ≥ 0.85 (black). Abbreviation: RQ—respiratory quotient; Th12 ESM_SMI_—skeletal muscle index of Th12 Erector Spinae muscles; VT—tidal volume; BMI—body mass index; %FEV_1_—FEV_1_ predicted.

**Figure 3 nutrients-14-02596-f003:**
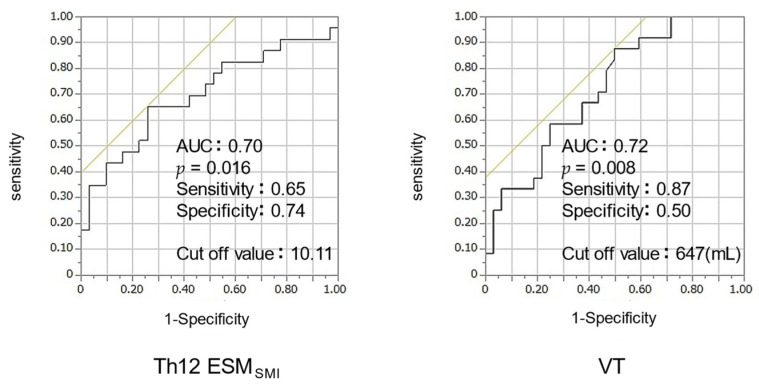
ROC analysis of Th12 ESM_SMI_, VT for energy malnutrition. Abbreviation: ROC—receiver operating characteristic, Th12 ESM_SMI_—skeletal muscle index of Th12 Erector Spinae muscles; VT—tidal volume.

**Table 1 nutrients-14-02596-t001:** Subject characteristics.

	Reference Value	Median (IQR)	Range (Min–Max)
Number (*n*)	N/A	56	N/A
Age (years)	N/A	74 (67–80)	60–93
Sex (female/male)	N/A	0/56	N/A
BMI (kg/m^2^)	18.5–24.9	22.5 (20.1–24.1)	15.0–27.2
%IBW (%)	N/A	102.4 (91.3–109.8)	68.2–123.6
GOLD stage (I/II/III/IV) (*n*)	N/A	12/23/18/3	N/A
mMRC dyspnea scale score (0/1/2/3/4) (*n*)	N/A	14/26/8/6/2	N/A
FEV_1_ (L)	N/A	1.59 (1.14–2.15)	0.47–2.81
%FEV_1_ (%)	N/A	58.8 (44.8–77.9)	18.9–110.6
FVC (L)	N/A	3.11 (2.56–3.70)	1.07–4.89
FEV_1_/FVC (%)	N/A	52.7 (42.5–63.8)	27.4–94.8
VC (L)	N/A	3.41 (2.91–3.86)	1.02–5.36
MNA-SF (point)	N/A	12 (10.2–13.0)	6–14
NR-ADL (point)	N/A	95 (90–97)	73–99
Albumin (g/dL)	4.1–5.1	4.2 (3.9–4.5)	3.1–4.8
Transthyretin (mg/dL)	22.0–40.0	28.3 (24.0–32.4)	8.7–44.9
CRP (mg/dL)	0.04<	0.09 (0.04–0.27)	0–6.4
AST (IU/L)	13–30	22 (18.2–26.0)	13–47
ALT (IU/L)	10–30	18 (15.0–23.5)	7–50
GGT (IU/L)	13–64	25.5 (18.0–34.7)	9–240
Total cholesterol (mg/dL)	142–219	197.5 (173–215)	110–251
HDL cholesterol (mg/dL)	40–86	63 (50.2–75.0)	36–111
LDL cholesterol (mg/dL)	60–139	117 (97.5–134)	46–173
Triglyceride (mg/dL)	40–149	97 (66.5–132.7)	44–266
Creatinine (mg/dL)	0.65–1.07	0.87 (0.75–1.03)	0.57–1.63
Blood glucose (mg/dL)	80–109	103 (95.2–114)	85–158
RQ	N/A	0.85 (0.81–0.88)	0.74–0.97
Carbohydrate oxidation rate (%)	N/A	50.8 (38.8–61.9)	10.9–90.5
Fat oxidation rate (%)	N/A	49.2 (38.5–61.1)	9.5–89.1

Note: data are expressed as median (interquartile range (IQR)), range, or number. Abbreviations: N/A—not applicable; BMI—body mass index; IBW—ideal body weight; GOLD—Global Initiative for Chronic Obstructive Lung Disease; MNA-SF—Mini Nutritional Assessment Short-Form; NR-ADL—Nagasaki university Respiratory ADL Questionnaire; CRP—C-reactive protein; AST—aspartate aminotransferase; ALT—alanine aminotransferase; GGT—gamma-glutamyl transpeptidase; RQ—respiratory quotient.

**Table 2 nutrients-14-02596-t002:** Comparison of baseline, energy metabolism, biochemical tests, body composition, physical function, and muscle mass in subjects with and without energy malnutrition (with RQ < 0.85 and RQ ≥ 0.85).

	Energy Malnutrition [RQ < 0.85] (*n* = 24)		Non = Energy Malnutrition [RQ ≥ 0.85] (*n* = 32)		
	Median (IQR)	Range (Min–Max)	Median (IQR)	Range (Min–Max)	*p*-Value
Age (years)	75 (65.2–81.0)	60–93	72 (67.5–77.7)	60–87	0.528
Sex (female/male)	0/24	N/A	0/32	N/A	N/A
BMI (kg/m^2^)	21.2 (19.6–23.6)	15.0–26.2	22.9 (20.6–24.9)	17.5–27.2	0.131
%IBW (%)	96.3 (89.1–107.5)	68.2–118.9	104.0 (93.9–113.3)	79.4–123.6	0.129
GOLD stage (I/II/III/IV) (*n*)	3/11/8/2	N/A	9/12/10/1	N/A	0.474
mMRC dyspnea scale score (0/1/2/3/4) (*n*)	6/8/3/5/2	N/A	8/18/5/1/0	N/A	0.080
FEV_1_ (L)	1.3 (1.1–2.1)	0.8–2.8	1.7 (1.2–2.2)	0.5–2.8	0.161
%FEV_1_ (%)	55.5 (44.1–71.3)	27.6–87.8	65.8 (46.5–81.1)	18.9–110.6	0.164
FVC (L)	2.9 (2.4–3.6)	1.1–3.8	3.2 (2.7–3.7)	1.5–4.9	0.179
FEV_1_/FVC (%)	49.9 (43.0–63.8)	27.4–87.8	54.7 (41.5–63.7)	30.6–94.8	0.746
VC (L)	3.3 (2.7–3.7)	1.0–4.3	3.5 (3.0–3.8)	1.7–5.3	0.376
REE (kcal/day)	1237 (1103–1421)	868–1531	1319 (1144–1375)	1040–1719	0.328
REE/kg (kcal/kg/day)	21.8 (19.3–23.7)	16.5–28.2	21.6 (19.5–22.7)	16.7–25.2	0.797
BEE (kcal/day)	1241 (1109–1424)	836–1714	1324 (1203–1473)	940–1705	0.138
REE/BEE	1.01 (0.90–1.10)	0.76–1.31	1.00 (0.90–1.05)	0.78–1.17	0.728
RQ	0.80 (0.77–0.83)	0.74–0.84	0.88 (0.86–0.89)	0.85–0.97	<0.001
Carbohydrate oxidation rate (%)	35.4 (23.3–43.6)	10.9–49.3	61.2 (53.0–67.1)	49.4–90.5	<0.001
Fat oxidation rate (%)	64.5 (56.3–76.6)	50.7–89.1	38.7 (32.8–46.9)	9.5–50.6	<0.001
RR (count/min)	14.1 (12.0–18.1)	8.0–23.6	13.8 (10.9–15.5)	6.6–20.6	0.236
VT (mL)	548 (474–634)	374–693	645 (552–708)	433–1105	0.005
VE (mL/min)	7.8 (6.6–8.7)	4.8–10.1	8.5 (7.3–9.0)	6.3–12.0	0.149
Albumin (g/dL)	4.2 (4.0–4.4)	3.1–4.7	4.3 (3.9–4.5)	3.4–4.8	0.796
Transthyretin (mg/dL)	28.4 (21.9–33.8)	8.7–44.9	28.2 (24.2–31.5)	20.0–44.2	0.953
CRP (mg/dL)	0.07 (0.03–0.44)	0–6.47	0.11 (0.04–0.27)	0–2.75	0.898
AST (IU/L)	21.0 (18.2–26.7)	13.0–46.0	22.5 (18.2–25.7)	14.0–47.0	0.721
ALT (IU/L)	18.0 (15.0–24.7)	7.0–47.0	18.0 (15.2–21.7)	9.0–50.0	0.993
GGT (IU/L)	24.5 (18.0–34.7)	14.0–240.0	27.0 (18.7–36.2)	9.0–216.0	0.888
Total cholesterol (mg/dL)	208 (181–215)	110–248	192 (170–220)	136–251	0.428
HDL cholesterol (mg/dL)	62 (51–72)	45–111	63 (50–77)	36–94	0.642
LDL cholesterol (mg/dL)	119 (97–137)	46–173	116 (91–133)	67–158	0.810
Triglyceride (mg/dL)	101 (66–130)	47–199	97 (66–137)	44–266	0.849
Creatinine (mg/dL)	0.89 (0.80–1.02)	0.59–1.63	0.85 (0.72–1.06)	0.57–1.39	0.562
Blood glucose (mg/dL)	97 (94–110)	86–149	106 (96–120)	85–158	0.095
Lean mass (kg)	45.5 (42.5–51.4)	35.8–56.5	47.3 (44.2–51.4)	35.7–58.8	0.278
SMI (BIA) (kg/m^2^)	7.5 (7.1–8.0)	6.7–8.4	7.4 (6.9–8.4)	5.8–9.0	0.961
Fat mass (kg)	11.5 (8.6–17.0)	2.6–23.2	12.7 (10.8–17.8)	4.2–25.3	0.190
Body fat percentage (%)	19.5 (16.3–26.4)	6.6–29.2	21.9 (18.5–26.6)	8.9–33.1	0.260
Upper arm circumference (cm)	23.1 (21.3–25.1)	18.3–28.8	24.3 (23.4–25.7)	19.7–29.0	0.072
Lower leg circumference (cm)	32.1 (28.9–33.2)	26.3–36.8	31.8 (30.4–35.2)	27.5–38.5	0.407
Grip strength (kg)	35.5 (30.1–40.2)	16.3–50.3	38.0 (32.1–41.6)	19.1–47.5	0.223
Knee extension strength (kgf)	27.9 (19.0–37.9)	14.8–45.1	33.1 (26.1–38.5)	18.0–52.8	0.060
Gait speed (m/s)	1.6 (1.4–2.0)	0.9–2.4	1.7 (1.5–1.8)	0.9–2.2	0.816
TUG (second)	6.9 (6.2–8.7)	5.3–13.1	7.0 (6.2–8.2)	5.4–11.7	0.772
CS-5 (second)	6.0 (5.3–8.3)	4.3–10.2	6.9 (6.1–8.8)	4.1–13.3	0.139
6MWT (m)	505 (450–540)	390–550	460 (427–480)	260–555	0.129
NR-ADL (point)	96.5 (91.7–98.0)	88–99	94.0 (87.5–97.0)	73–98	0.062
MNA-SF (point)	11 (10–12)	6–14	13 (11–13)	7–14	0.013
Th12 muscle mass (cm^2^)	50.1 (43.2–58.7)	25.3–87.4	53.9 (48.2–65.1)	35.1–75.1	0.046
Th12 SMI (cm^2^/m^2^)	17.9 (15.5–21.5)	10.0–31.9	21.1 (17.8–23.3)	12.8–27.4	0.050
Th12 ESM muscle mass (cm^2^)	27.2 (19.8–31.2)	8.8–45.2	31.5 (27.4–35.9)	19.8–40.5	0.014
Th12 ESM_SMI_ (cm^2^/m^2^)	9.9 (7.9–11.4)	3.6–16.5	11.4 (9.9–12.8)	6.8–15.0	0.013

Note: data are expressed as median (interquartile range (IQR)), range, or number. Abbreviations: BMI—body mass index; IBW—ideal body weight; GOLD—Global Initiative for Chronic Obstructive Lung Disease; REE—resting energy expenditure; BEE—basal energy expenditure; RQ—respiratory quotient; RR—respiratory rate; VT—tidal volume; VE—minute ventilation; CRP—C-reactive protein; AST—aspartate aminotransferase; ALT—alanine aminotransferase; GGT—gamma-glutamyl transpeptidase; SMI—skeletal muscle index; TUG—timed up and go test; CS-5—5-time chair stand test; 6MWT—6-minute walk test; MNA-SF—Mini Nutritional Assessment Short-Form; NR-ADL—Nagasaki university Respiratory ADL Questionnaire; ESM—Erector Spinae muscles; Th12 ESM_SMI_—skeletal muscle index of Th12 Erector Spinae muscles. *p*-values are for the comparison of groups by the Wilcoxon rank-sum test.

**Table 3 nutrients-14-02596-t003:** Logistic regression analysis for energy malnutrition (RQ < 0.85).

Factors	Odds Ratio	95% Confidence Interval	*p*-Value
Th12 ESM_SMI_ (cm^2^/m^2^)	0.73	0.517–0.978	0.034
VT (mL)	0.99	0.984–0.998	0.010
MNA-SF (point)	0.91	0.578–1.398	0.676

Abbreviations: Th12 ESM_SMI_—skeletal muscle index of Th12 Erector Spinae muscles; VT—tidal volume; MNA-SF—Mini Nutritional Assessment Short-Form.

## Data Availability

All data generated or analyzed during this study are included in this article. Further inquiries can be directed to the corresponding author.

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
