# Peer review of "The Relationship of Energy Malnutrition, Skeletal Muscle and Physical Functional Performance in Patients with Stable Chronic Obstructive Pulmonary Disease"

_nutrients, 2022, doi:10.3390/nu14132596_

Round 1
Reviewer 1 Report
1. The authors of this paper has collected a large number of clinical indicators for analysis, which has guiding significance for clinical practice. However, in terms of statistical methods, why not carry out single factor regression analysis first, and then carry out multi factor regression analysis?
2. A small question about the measurement of RQ. It is mentioned in the article that: "All measurements were taken at 9:30 a.m. in a fasted state", do patients collect metrics the next morning after a one-night stay in the hospital, or they travel from home to the hospital the next morning then collect the metrics? If it is the latter, will the movement of going to the hospital cause changes in the metabolic level and affect the measurement results of RQ due to the different distances between different patients' places of residence and the hospital and the different ways of going to the hospital? For example, the metabolic and respiratory function consumption of a patient living 1km away from the measurement point walking to the hospital is different from that of a patient living 1km away from the measurement point taking public transport to the hospital. Does this possible "movement" before measurement affect the patient's RQ?
3. The innovation of this study and its guiding significance for clinical work can be further expanded.
Reviewer 2 Report
your study evaluates energy-malnutrition or protein energy undernutrition (PEU) in patients with COPD and its determinants
the purpose of the study is very interesting and useful in this setting but I have the following observations:
1. study design: you mention in the text that you enrolled consecutive patients and this is not compatible with the cross-sectional design claimed in the beginning of the methodology. So that I recommend you to mention that this is a prospective observational study
2. in the methodology study please describe briefly why you used that battery of blood tests
3. you found VT as one of the determinants of energy malnutrition. could you please briefly explain how you secured that this lower VT is only due to respiratory muscle sarcopenia and not to static/dynamic hyperinflation?
4 in the methodology section please describe how the lung function variables were measured. (briefly)
Round 2
Reviewer 1 Report
Based on previous comments, the author has updated and improved the manuscript. Here I would like to make another suggestion about the title of the article. Firstly, this study was a cross-sectional study, which applies to illustrating “correlation” rather than “causation” or “influence”. In addition, reading through the manuscript, I suppose the article(including the “Abstract”, “Result”, “Discussion” sections) focused on the risk factors of energy malnutrition or the association between some identified factors and energy malnutrition, which also make up the most important part of the conclusion, aiming at early detection and prevention of malnutrition. To sum up, the present title which means “the influence of energy malnutrition on other variables” might need to be reconsidered.
